# RIG-I Plays a Dominant Role in the Induction of Transcriptional Changes in Zika Virus-Infected Cells, which Protect from Virus-Induced Cell Death

**DOI:** 10.3390/cells9061476

**Published:** 2020-06-16

**Authors:** Mirjam Schilling, Anne Bridgeman, Nicki Gray, Jonny Hertzog, Philip Hublitz, Alain Kohl, Jan Rehwinkel

**Affiliations:** 1Medical Research Council Human Immunology Unit, Medical Research Council Weatherall Institute of Molecular Medicine, Radcliffe Department of Medicine, University of Oxford, Oxford OX3 9DS, UK; mirjam.schilling@ndm.ox.ac.uk (M.S.); anne.bridgeman@imm.ox.ac.uk (A.B.); jonny.hertzog@kellogg.ox.ac.uk (J.H.); 2MRC WIMM Centre for Computational Biology, MRC Weatherall Institute of Molecular Medicine, University of Oxford, Oxford OX3 9DS, UK; nicki.gray@imm.ox.ac.uk; 3Genome Engineering Facility, Medical Research Council Weatherall Institute of Molecular Medicine, Radcliffe Department of Medicine, University of Oxford, John Radcliffe Hospital, Oxford OX3 9DS, UK; philip.hublitz@ndcls.ox.ac.uk; 4MRC-Centre for Virus Research, University of Glasgow, Glasgow G61 1QH, UK; Alain.Kohl@glasgow.ac.uk

**Keywords:** Zika virus, IFN, RIG-I, MDA5, apoptosis, NS5, IFNAR1

## Abstract

The Zika virus (ZIKV) has received much attention due to an alarming increase in cases of neurological disorders including congenital Zika syndrome associated with infection. To date, there is no effective treatment available. An immediate response by the innate immune system is crucial for effective control of the virus. Using CRISPR/Cas9-mediated knockouts in A549 cells, we investigated the individual contributions of the RIG-I-like receptors MDA5 and RIG-I to ZIKV sensing and control of this virus by using a Brazilian ZIKV strain. We show that RIG-I is the main sensor for ZIKV in A549 cells. Surprisingly, we observed that loss of RIG-I and consecutive type I interferon (IFN) production led to virus-induced apoptosis. ZIKV non-structural protein NS5 was reported to interfere with type I IFN receptor signaling. Additionally, we show that ZIKV NS5 inhibits type I IFN induction. Overall, our study highlights the importance of RIG-I-dependent ZIKV sensing for the prevention of virus-induced cell death and shows that NS5 inhibits the production of type I IFN.

## 1. Introduction

The recent epidemic caused by Zika virus (ZIKV), a mosquito-borne flavivirus, revealed the potential of the virus to inflict severe harm on infected individuals. Whereas infection is often asymptomatic or causes a self-limiting acute febrile illness in adults, it has been linked to multiple neurodevelopmental defects, including microcephaly, in newborns [1,2]. ZIKV has furthermore been associated with cases of Guillain–Barré syndrome, an autoimmune disease resulting in rapid-onset muscle weakness [3,4].

Experiments in cell culture and mice showed that ZIKV infection is controlled by type I interferons (IFNs), which represent the first line of defense against viral infections [5,6,7,8]. Type I IFNs are induced by pattern recognition receptors (PRRs) upon sensing of pathogen associated molecular patterns. For example, viral RNAs are detected by RIG-I-like receptors (RLRs) including retinoic acid-inducible gene I (RIG-I) and melanoma differentiation-associated gene 5 (MDA5). These PRRs then activate mitochondrial antiviral-signaling protein (MAVS). Downstream of MAVS, transcription factors such as interferon regulatory factor 3 (IRF3) induce the expression and secretion of type I IFNs [9]. Binding of the secreted type I IFNs to their receptor in turn induces a signaling cascade involving signal transducer and activator of transcription (STAT) 1 and 2 hetero-dimerization that leads to the expression of interferon stimulated genes (ISGs). Some ISGs encode antiviral proteins that then restrict viral replication. Deficiency of PRRs increases ZIKV replication in human skin fibroblasts and mice lacking the type I IFN receptor (IFNAR), STAT2, MAVS or a combination of IRF transcription factors show higher viral replication and pathology [5,7,8,10,11,12]. ZIKV induces type I IFNs, and this is mainly dependent on MAVS, suggesting an important role of RIG-I and/or MDA5 in virus detection [7,13,14,15,16,17,18]. ZIKV antagonizes IFNAR signaling as viral replication is blocked only moderately if type I IFN is applied with or after ZIKV infection [19]. NS5, the RNA-dependent RNA polymerase that replicates the viral genome, is a potent viral antagonist of IFNAR signaling [6,13,20]. NS5 proteins from African and French Polynesian isolates were reported to interact with and target STAT2 for degradation [6,21]. Furthermore, NS5 prevents STAT1 and STAT2 phosphorylation [19,22,23].

Here, we explored the contribution of RIG-I and MDA5 to ZIKV sensing and found that RIG-I was the main sensor required to induce a protective type I IFN response upon virus infection. Loss of RIG-I-mediated type I IFN production in infected A549 cells led to the activation of apoptosis. We furthermore show that ZIKV NS5 not only interferes with IFNAR signaling, but additionally inhibited type I IFN induction upstream of type I IFN gene transcription.

## 2. Materials and Methods

### 2.1. Cell Lines

A549 cells (kind gift from G. Kochs, Freiburg), HEK293 and Vero cells (kind gifts from C. Reis e Sousa, London, UK) and A549 BVDV NPro cells (kind gift from R. Randall, St Andrews) were cultured at 37 °C in Dulbecco’s modified Eagle’s medium (DMEM), supplemented with 10% FCS and 2-mM L-Glutamine.

### 2.2. Generation of Knock-Out Cells

To stably knock out RIG-I and MDA5, sgRNAs cloned into pX458-Ruby (Addgene 110164, deposited by Dr. Philip Hublitz) and described earlier [13] were used. To stably knock out IFNAR1, a sgRNA targeting *IFNAR1* exon 3 was selected based on the MIT algorithm (crispr.mit.edu) and cloned into pX458 (Addgene 48138, deposited by Dr. Feng Zhang). A549 and HEK293 cells were single-cell FACS sorted according to the co-expressed fluorescent protein (Ruby^+^ for cells transfected with the sgRNAs targeting RIG-I or MDA5, GFP^+^ for IFNAR1) 48 h post transfection. After 4 weeks, cells that had grown out to confluency were subjected to cell line characterization. We extracted genomic DNA and analyzed the target locus with a PCR screening protocol using primers up- and downstream of the sgRNA target sites. Primer sequences were: RIG-I (fwd: ttacattgtctcagactaagaggc, rev: gtgaagaatgggcacagtcggcc), MDA5 (fwd: cgtcattgtcaggcacagag, rev: agctctgccactgtttttcc) and IFNAR (fwd: gtgtatgctaaaatgttaatagg, rev: cctttgcgaaatggtgtaaatgag). Full knock-out was verified by submission of sequencing reads to TIDE (https://tide.nki.nl), an algorithm that decomposes sequencing data and allows determination of the spectrum of indels and their respective frequencies. Additionally, whole cell lysates were analyzed by western blot after stimulation with recombinant type I IFN (IFN-A/D, Sigma, 100 U/mL).

### 2.3. ZIKV

The Brazilian ZIKV isolate ZIKV/*H. sapiens*/Brazil/PE243/2015 was originally described in [24] and was grown on Vero cells. Viral titers were determined by plaque assay on A549 BVDV NPro cells. These cells are optimized for virus growth as they stably express the NPro protein of bovine viral diarrhea virus (BVDV), which induces degradation of IRF3 [25].

### 2.4. IFNβ ELISA

Human IFNβ concentrations in cell culture supernatants were measured by enzyme-linked immunosorbent assay (ELISA) using LumiKine™ Xpress hIFN-β 2.0 (Invivogen, cat. nb.: luex-hifnbv2) according to the manufacturer’s instructions.

### 2.5. Type I IFN Bioassay

To measure type I IFN bioactivity, HEK293 cells stably expressing the pGF1-ISRE reporter [26]—in which firefly luciferase expression is driven by interferon-stimulated response elements—were incubated with untreated supernatants of infected cells. Relative light units (RLUs) were measured with a luminometer after 24 h using ONE-Glo Luciferase Assay System (Promega) according to the manufacturer’s instructions and OptiPlate^TM^ 96-well plates (Perkin Elmer, cat. nb.: 6005299).

### 2.6. IFNβ Promoter Reporter Assay

HEK293 cells seeded in 96-well plates were transiently transfected with 50 ng of ZIKV-NS3 or -NS5 expression plasmid, 20 ng of a plasmid encoding firefly luciferase (F-Luc) under the control of the *IFNβ* promoter and 5 ng pRL-TK, a plasmid which constitutively expresses renilla luciferase (R-Luc). Twenty-four hours later, cells were transfected with 5 ng IVT–RNA or 50 ng Hela–EMCV–RNA per well [13]. F-Luc activity was determined 24 h after RNA transfection using Dual-Luciferase Reporter Assay System (Promega) and normalized to R-Luc activity.

### 2.7. Caspase Activity Assay

Caspase 3/7 Glo assay (Promega) was performed according to the manufacturer’s instructions.

### 2.8. qRT-PCR

Cells were lysed and total RNA was extracted using the QIAshredder (Qiagen) and RNeasy Mini Kit (Qiagen) according to the manufacturer’s instructions. RNA was reverse transcribed using SuperScript II Reverse Transcriptase (Invitrogen) into cDNA that was then used for qPCR with either TaqMan Universal PCR Master Mix (Applied Biosystems) or SYBR green PCR kit (Life Technologies). *C_T_* values were normalized to GAPDH (Δ*C_T_*). TaqMan primer probes used include GAPDH (Assay ID: Hs02758991 g1), IFIT1 (Assay ID: Hs03027069 s1) and MX1 (Assay ID: Hs00895608 m1). SYBR green primer probes used include GAPDH (fwd: CATGGCCTTCCGTGTTCCTA, rev: CCTGCTTCACCACCTTCTTGA) and ZIKV (fwd: CGAGGAACATCCAGACTC, rev: ATTGGAGATCCTGAAGTTCC).

### 2.9. 3′mRNA Sequencing

A549 wt, RIG-I KO (clone B05) and MDA5 KO (clone c27) cells were lysed at 24 h post infection with ZIKV (multiplicity of infection (MOI): 5), and total RNA was extracted using the QIAshredder (Qiagen) and RNeasy Mini Kit (Qiagen) according to manufacturer’s instructions. RNA concentration was measured by the Qubit RNA HS (high sensitivity) Assay Kit (Life Technologies) according to the manufacturer’s instructions. Quality of the extracted RNA was controlled by the Agilent 2200 TapeStation System (Agilent Technologies).

cDNA was generated from total RNA using first oligo-dT and subsequently random priming. The prepared libraries were QC’ed and multiplexed before sequencing over one lane of the NextSeq flow cell (high output, 75 bp single reads).

Following QC analysis with the fastQC package, reads were aligned using STAR against the human genome assembly (GRCh38 (hg38) UCSC transcripts) [27]. Read counts were visualized using UCSC genome browser [28]. Gene expression levels were quantified as read counts using the featureCounts function from the Subread package with default parameters [29]. The read counts were used for the identification of global differential gene expression between specified populations using the edgeR package [30]. RPKM values were also generated using the edgeR package [30]. Genes were considered differentially expressed between populations if they had an adjusted *p*-value (false discovery rate, FDR) of less than 0.05. The edgeR package was also used to generate heatmaps and plots [30]. The Venn diagram was created using [31].

### 2.10. FACS

HEK293 cells were trypsinized, resuspended in FACS buffer (PBS, 1% FCS, 2 mM EDTA, 0.02% sodium azide) and fixed immediately by adding an equal volume of pre-warmed (to 37 °C) BD Cellfix (BD Biosciences). Cells were permeabilized by adding chilled BD Phosflow Perm Buffer III (BD Biosciences) drop-by-drop while vortexing. Intracellular staining was performed with anti-p-STAT1 antibody (mouse anti-human p-STAT1, BD clone 4a (RUO)). Cellular debris and doublets were gated out using forward scatter and side scatter channels and 50,000 live single cells were analyzed per sample using an Attune Flow Cytometer (ThermoFisher Scientific).

### 2.11. Western Blot Analysis

Cells were lysed in YG-lysis buffer (10 mM Trizma, 50 mM NaCl, 30 mM sodium pyrophosphate, 5 mM sodium fluoride, 5 μM ZnCl_2_, 10% NP40, plus protease inhibitor) and the samples were incubated at 95 °C for 5 min. Protein lysates were separated on 10% SDS-PAGE gels and transferred onto polyvinylidene difluoride (PVDF) membranes. As primary antibodies we used anti-MDA5 (mouse, [13]), anti-Mx (mouse; M143 [32]), anti-PARP (rabbit, cell signaling), anti-RIG-I (mouse, AdipoGen), anti-IRF3 (D6I4C, rabbit, cell signaling), anti-p-IRF3-S396 (4D4G, rabbit, cell signaling), anti-ZIKV NS3 and anti-ZIKV NS5 sera (kind gift from A. Merits, Tartu) and anti-beta-actin-HRP (clone AC-15, Sigma-Aldrich). Detection of the primary antibodies was performed by the use of peroxidase-conjugated secondary antibodies (GE Healthcare).

## 3. Results

### 3.1. RIG-I Is the Main Sensor for ZIKV Infection in A549 Cells

Previously, we showed that total RNA extracted from cells infected with ZIKV contains immunostimulatory RNAs that activate the MAVS pathway when transfected into reporter cells [13]. To study the contribution of RIG-I and MDA5 to sensing of live ZIKV in infected cells, we screened a panel of cell lines for type I IFN induction after ZIKV infection. We found that the lung adenocarcinoma cell line A549 produced robust levels of type I IFNs upon infection (data not shown). Furthermore, this cell line was well established for studies of type I IFN responses and was therefore chosen for this work. We generated A549 cells lacking either RIG-I or MDA5 by CRISPR/Cas9-mediated knock-out (KO) and validated these by western blot analysis and sequencing (Figure 1A,B). We obtained one MDA5 KO clone designated c27. MDA5 protein expression was undetectable in these cells (Figure 1A). Sequencing of the region targeted by the sgRNA suggested a +1 insertion on one allele and a −5 deletion on the other, both of which disrupt the reading frame (Figure 1B). Three RIG-I KO clones were generated, and all had no detectable RIG-I protein (Figure 1A). RIG-I KO clone B05 was used in subsequent experiments; sequencing showed a +1 insertion as well as −1 and −2 deletions at the sgRNA target site, which all disrupt the reading frame (Figure 1B). The presence of three different alleles could be explained by the triploidy of much of the A549 genome [33,34].

In order to compare the amounts of type I IFN produced, wild-type (wt) and KO A549 cells were infected with ZIKV using a multiplicity of infection (MOI) of 0.1 or 1. After 24 h, we collected supernatants and measured IFNβ levels by ELISA. These virus doses and the timepoint were chosen to monitor type I IFN responses to incoming virus early after infection. Similar amounts of IFNβ were present in supernatants from wt and MDA5 KO cells (Figure 1C). In contrast, little or no IFNβ was detectable in samples from RIG-I KO cells. Next, we measured bioactive type I IFN levels in supernatants collected from cells infected (MOI 1) for 48 h by using a bioassay: supernatant samples were transferred onto HEK293 cells with a stably integrated pGF1-ISRE reporter [26]. These cells harbor an F-Luc gene under control of interferon-stimulated response elements (ISREs) that were bound and activated by STAT1/2 upon engagement of IFNAR. Cells stimulated with the supernatant of infected wt or MDA5 KO cells induced similar amounts of F-Luc, whereas the supernatant of infected RIG-I KO cells did not lead to significant F-Luc induction (Figure 1D). Furthermore, we tested the activation of IRF3 in infected cells by western blot using an antibody recognizing S396-phosphorylated IRF3 (p-IRF3). This analysis revealed IRF3 phosphorylation upon ZIKV infection in wt and MDA5 KO cells, but not in RIG-I KO cells (Figure 1E). At the selected MOIs and 24-h timepoint analyzed, infection levels were similar in cells of all genotypes as indicated by comparable levels of the viral NS3 protein (Figure 1E). In summary, these data demonstrated that loss of RIG-I abrogated the induction and secretion of type I IFN in A549 cells upon ZIKV infection. To examine the impact of reduced IRF3 activation and type I IFN secretion on ISG induction, A549 cells were infected with ZIKV (MOI 1 or 5) and *IFIT1* and *MX1* mRNA levels were quantified by RT-qPCR. *IFIT1* mRNA was robustly induced in A549 wt and MDA5 KO cells, whereas no induction was detectable in A549 RIG-I KO cells (Figure 1F). Similarly, induction of *MX1* transcripts was not detectable in RIG-I KO cells; however, in contrast to *IFIT1*, *MX1* induction was also reduced in MDA5 KO cells (Figure 1F). This suggested that a subset of ISGs was controlled by both RIG-I and MDA5. To determine the impact of individual RLRs on ZIKV replication in a setting where the infection spreads between cells, we infected cells with a low dose of ZIKV (MOI 0.1) and ZIKV RNA was quantified by RT-qPCR up to 5 days post infection. Virus replication was similar in A549 wt and MDA5 KO cells; however, the virus replicated more potently from day 3 onwards in RIG-I KO cells (Figure 1G). Taken together, these data suggest that RIG-I was the main sensor that detects ZIKV infection in A549 cells leading to the induction of type I IFNs and ISGs. In turn, absence of RIG-I facilitated virus replication.

### 3.2. Transcriptomic Analysis of ZIKV-Infected Cells Indicates that RIG-I Plays a Dominant Role in ISG Induction

In light of our observation that the induction of *MX1* transcripts after ZIKV infection was not only RIG-I, but also partially MDA5-dependent, we wanted to further investigate how the individual receptors influence transcriptomic changes after virus infection. We therefore performed 3‘ mRNA sequencing of total RNA that was extracted from ZIKV-infected cells 24 h after infection using an MOI of 5 to robustly induce ISGs. Our analysis included four biologic replicates each for uninfected and infected wt, RIG-I KO or MDA5 KO cells. Across cells of all genotypes, a total of 236 genes were differentially regulated upon ZIKV infection (Figure 2A). Most genes were differentially expressed in A549 wt and MDA5 KO cells with a substantial overlap between the two. Only 24 genes were differentially regulated in RIG-I KO cells, indicating that transcriptomic changes upon ZIKV infection were largely driven by RIG-I. This was also evident from the heatmap in Figure 2B where ZIKV-infected RIG-I KO cells rather clustered with the uninfected instead of the ZIKV-infected cells of other genotypes. Most the 236 genes differentially expressed upon ZIKV infection were upregulated while only few were downregulated (Figure 2B). Next, we analyzed ISGs using the gene set defined in [35]. A total of 98 genes differentially expressed in ZIKV-infected cells were ISGs. Most ISGs—such as *IFIT1* or *RSAD2* (also known as viperin)—were upregulated in a RIG-I-dependent and MDA5-independent manner (Figure 2C,D). Induction of a small number of ISGs not only required RIG-I but was also partially MDA5-dependent (Figure 2C). As predicted from our RT-qPCR results shown in Figure 1D, this included *MX1* (Figure 2D). Other ISGs were induced to a similar extent upon infection in all three cell lines, including *CCL5* (Figure 2C,D). RIG-I and MDA5 may be redundant for activation of these genes—or their induction could require other signaling pathways. Taken together, these data show that most transcriptional changes in A549 cells upon ZIKV infection occurred in a RIG-I-dependent manner and that RIG-I was particularly important for the induction of ISGs.

### 3.3. RIG-I-Mediated Signaling Protects A549 Cells from Apoptosis

ZIKV infection causes apoptosis [36,37,38,39,40,41,42,43]. We therefore asked whether reduced innate immune recognition of ZIKV in RIG-I KO cells impacts virus-induced cell death. A549 wt, RIG-I KO and MDA5 KO cells were infected with a low dose of ZIKV (MOI 0.1) to analyze a spreading infection and the confluency of the cells was measured for 6 days using an in-incubator imaging system (Incucyte). Interestingly, we found that after 6 days of infection the confluency of RIG-I KO cells was decreased by about 50%, whereas the confluency of wt and MDA5 KO cells was not affected by ZIKV infection (Figure 3A). Furthermore, crystal violet staining revealed virus-induced cell death in two different RIG-I KO clones, but not in wt or MDA5 KO cells six days after infection with two different doses of virus (MOI 0.1 and 0.01) (Figure 3B). To determine whether apoptosis was induced during ZIKV infection in RIG-I KO cells, we performed western blot analysis of PARP cleavage, a molecular signature of apoptosis. Indeed, ZIKV infection resulted in increased levels of cleaved PARP in RIG-I KO cells, but not in wt and MDA5 KO cells (Figure 3C). In addition, we monitored activity of the apoptotic caspases-3 and -7. Four days after ZIKV infection, RIG-I KO cells showed a 4-fold induction of caspase-3/7 activity, while only a 2-fold induction was observed in wt and MDA5 KO cells (Figure 3D). Furthermore, ZIKV-infected cells displayed shrinkage and membrane blebbing, morphologic changes typical for apoptotic cells (Figure 3E). Taken together, these data showed that a lack of RIG-I signaling in A549 cells led to a loss of protection from ZIKV-induced apoptosis, which may be due to reduced type I IFN production and increased virus replication in RIG-I-deficient cells.

### 3.4. ZIKV NS5 Inhibits Type I IFN Induction

ZIKV NS5 inhibits type I IFN signaling by inducing degradation of STAT2 and by blocking phosphorylation of STAT1 [6,13,19,21,44]. Results of overexpression studies suggested that NS5 also blocks the innate immune response by inhibiting the induction of type I IFN [6,13,44,45]. However, it is possible that the latter effect is indirect as RLRs and many proteins involved in their downstream signaling are encoded by ISGs [46]. As such, lower levels of type I IFN induction in cells expressing NS5 could be explained by reduced levels of RLRs or other proteins involved in type I IFN induction. To distinguish between such indirect effects of NS5 and direct inhibition of type I IFN induction, we generated HEK293 cells lacking IFNAR1 and obtained one clone designated c27. As expected, IFNAR1 KO cells were incapable to phosphorylate STAT1 in response to IFNα2a (Figure 4A). We further validated our IFNAR1 KO cells by sequencing and found −1, −3 and −5 deletions at the sgRNA target site (Figure 4B; HEK293 are largely triploid [47,48]). While the −1 and −5 deletions result in a frameshift, the −3 deletion removes one amino acid keeping the reading frame intact. Our functional data in Figure 4A showing the absence of response to IFNα2a suggest that the protein encoded by the -3 mutant allele is either non-functional or rapidly degraded.

We then transfected wt and IFNAR1 KO HEK293 cells with an expression plasmid for ZIKV NS5. We used empty vector and ZIKV NS3, which we previously found not to block RLR signaling [13] or EMCV L, which blocks IRF3 [49], as negative and positive controls, respectively. Alongside these expression plasmids, cells were co-transfected with an *IFNβ* promoter F-Luc reporter construct and R-Luc as a transfection control. Next, we stimulated RIG-I or MDA5 by transfecting 5′-triphosphate containing in vitro transcribed RNA (IVT–RNA) or RNA extracted from EMCV-infected Hela cells (Hela–EMCV–RNA), respectively (Figure 4C, [13]). Luciferase activities were measured 24 h after RNA transfection. As expected, both immunostimulatory RNAs induced the *IFNβ* promoter in wt cells that had been transfected with empty vector or ZIKV NS3, while the response was strongly reduced by ZIKV NS5 and EMCV L (Figure 4D). Importantly, NS5 inhibited induction of the *IFNβ* promoter to a similar extent in IFNAR1 KO cells stimulated with IVT–RNA (Figure 4D). In addition, the response to Hela–EMCV–RNA also appeared to be reduced by NS5 in IFNAR1 KO cells, although this trend did not reach statistical significance. We conclude that ZIKV NS5 blocked RIG-I-mediated IFN induction in the absence of IFNAR signaling. These observations suggest that ZIKV NS5 not only inhibits antiviral responses downstream of IFNAR signaling, but also has a direct effect on the induction of type I IFNs by RLRs.

## 4. Discussion

Our data demonstrate that RIG-I is the main sensor for ZIKV infection in A549 cells. Genetic ablation of RIG-I in A549 cells led to a loss of type I IFN production and ISG induction as well as to an increase in virus titer. Transcriptomic analysis corroborated that knockdown of RIG-I strongly reduced differential gene expression upon ZIKV infection. Importantly, most ISGs induced after ZIKV infection were RIG-I-dependent. This is likely due to a combination of reduced type I IFN secretion by RIG-I-deficient cells and reduced activation of IRF3, which directly regulates some ISGs [46]. This RIG-I-dependency of ZIKV sensing is in line with a recent publication by Esser-Nobis and colleagues [16]. RIG-I is thought to recognize the conserved 5′-triphosphate group found on nascent RNAs of flaviviruses as shown by Chazal et al. [50]. This is interesting as flaviviruses replicate in complexes formed in invaginations of ER membranes, raising the question as to how RIG-I gains access to viral RNAs [51]. Replication factories are likely to be dynamic structures and viral RNAs can potentially leak into the cytoplasm. Studies on dengue virus, West Nile virus and tick-borne encephalitis virus furthermore revealed 10-nm-wide openings of these invaginations to the cytoplasm using electron tomography [51]. ZIKV RNA is thought to exit to the cytoplasm through these pores to be packaged into virions and for protein translation [52]. RIG-I can be activated by less than 20 RNA molecules per cell [53]. A few nascent positive-stranded RNA molecules that escape replication factories would therefore be sufficient to induce an IFN response by RIG-I.

Our study furthermore revealed that RIG-I prevents ZIKV-induced apoptosis, likely due to RIG-I-induced innate immunity that curtails virus replication. ZIKV-infected A549 RIG-I KO cells succumbed to cell death 4 days post infection with a low MOI and showed increased cleavage of PARP as well as activation of caspases-3 and -7. Several studies suggested that ZIKV-induced cell death in neuronal cells is responsible for neurodevelopmental defects such as microcephaly [36,43,54]. An increase in cell death upon ZIKV infection was modeled in brain-specific organoids derived from human induced pluripotent stem cells (iPSCs) [55,56]. Studies in different cell lines suggest that the induction of apoptosis may be cell-type specific. A549 cells were shown to succumb to apoptosis 48 h after infection when infected with a high MOI [57]. In contrast, human monocyte-derived dendritic cells, Vero cells or mosquito C6/36 cells infected with several different African or Asian ZIKV strains did not induce apoptosis 24 and 48 h after infection [58]. Our work now shows that a functional immune response to ZIKV infection protects cells from apoptosis. The observed cell-type specific differences could therefore correlate with how efficiently a cell senses the virus and how potently the virus is restricted by the initiated type I IFN response. It is thus important to study the levels and functionality of RIG-I in cell types infected by ZIKV, including neuronal cells. Furthermore, ZIKV delays apoptosis by modulating the activities of anti-apoptotic Bcl-2 family proteins [59]. It will be interesting for future studies to determine if and how viral targeting of RIG-I and Bcl-2 proteins is functionally linked.

The importance of RIG-I and type I IFN to ZIKV infection is also evident from the presence of viral antagonists. Here, we confirmed that NS5—one of the most potent viral antagonists—not only blocks type I IFN signaling, but also efficiently and directly inhibited type I IFN production triggered by RIG-I, as suggested previously [6,13,44,45,60]. It is now important to identify the precise mechanisms by which the RLR signaling cascade is targeted by NS5. Recently, Li et al. described that NS5 directly represses K63-linked polyubiquitination of RIG-I [61]. In addition, an inhibitory effect of NS5 on IRF3 activation has been suggested [44,62,63]. Interestingly, Lin and colleagues reported an interaction of NS5 with TBK1 that results in reduced phosphorylation of IRF3 [45]. The latter findings were made by NS5 overexpression in HEK293 cells. It would be very interesting in future studies to confirm these findings at endogenous protein level during infection with live ZIKV. In ZIKV-infected cells, RIG-I-dependent responses are induced despite the presence of NS5 (Figure 1, Figure 2 and Figure 3). It is therefore likely that NS5′s ability to block RIG-I signaling is not absolute. Whether this relates to relative protein levels in infected cells or to cell-to-cell variability remains to be determined. A virus expressing an NS5 mutant that fails to interfere with RIG-I signaling but maintains other functions of NS5 would be useful for such studies. It is noteworthy that a recent report using immortalized human fetal astrocytes and an siRNA approach found that both RIG-I and MDA5 were required for induction of *IFNβ* and *ISG* transcripts [17]. This work further described an inhibitory effect of ZIKV NS3 on RIG-I and MDA5 signaling, in apparent contradiction to our data shown here in Figure 4D and in [13]. It is possible that these differences relate to cell-type specific expression of co-factors of RIG-I and MDA5, an interesting hypothesis for future studies.

Taken together, our study emphasizes the importance of RIG-I-mediated-ZIKV-sensing in controlling virus replication and virus-induced cell death. Targeting viral antagonists to support ZIKV sensing by RIG-I may open up novel treatment options and limit the severity of ZIKV associated neurological symptoms.

## Figures and Tables

**Figure 1 cells-09-01476-f001:**
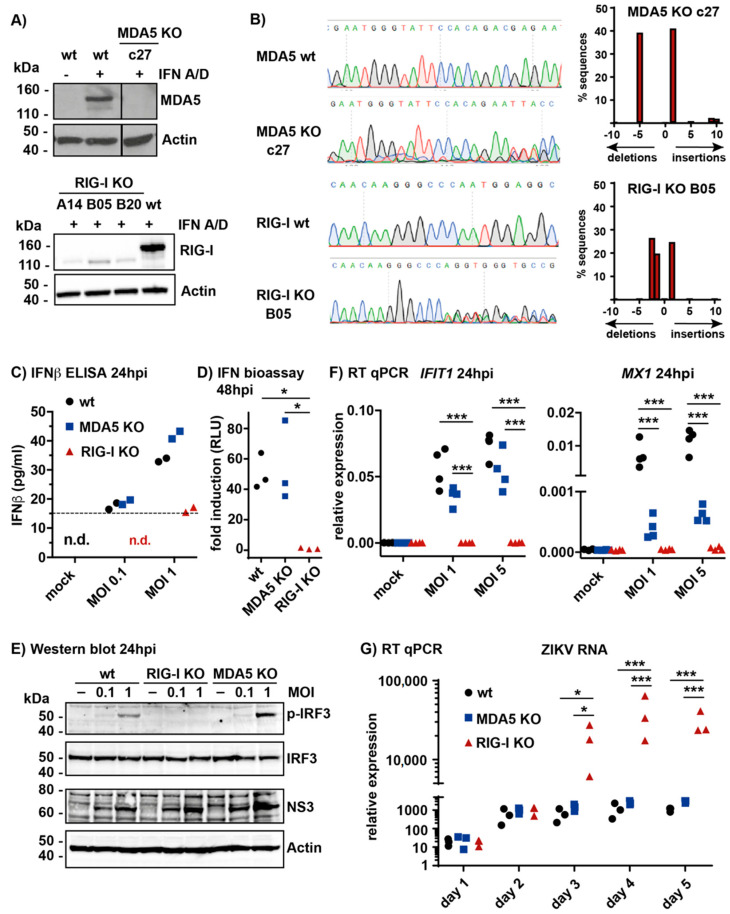
Retinoic acid-inducible gene I (RIG-I) is the main sensor for Zika virus (ZIKV) infection in A549 cells. (**A**) A549 cells were knocked-out for melanoma differentiation-associated gene 5 (MDA5) or RIG-I as described in Materials and Methods. Cells were stimulated with recombinant type I IFN (IFN-A/D) to induce MDA5 and RIG-I to detectable levels and lysates were analyzed by western blot using the indicated antibodies. Actin served as a loading control. The vertical line indicates a cut combining two parts of the same blot. c27, A14, B05 and B20 are individual clones. wt, wild type; (**B**) Genomic DNA was extracted from MDA5 KO clone c27 or RIG-I KO clone B05 cells. A fragment of DNA surrounding the targeted area was amplified by PCR and sequenced (left). Sequences were analyzed using TIDE (right). The number of nucleotides inserted or deleted, and the percentage of sequences affected are shown; (**C**) A549 MDA5 KO or RIG-I KO cells (clone B05) were infected with ZIKV (MOI 0.1 or 1), supernatant was collected 24 h later and IFNβ levels were analyzed by ELISA. The horizontal dashed line indicates the detection limit; n.d., not detectable; (**D**) A549 cells were infected with ZIKV (MOI 1) and supernatant was collected 48 h later. HEK293 cells stably expressing the pGF1-ISRE reporter were incubated with the supernatant and F-Luc activity was measured after 24 h. Shown is the fold induction relative to supernatant from mock infected cells. (**E**) A549 cells were infected with ZIKV (MOI 0.1 or 1), protein samples were collected 24 h later and analyzed by western blot using the indicated antibodies. Actin served as a loading control; (**F**) A549 cells were infected with ZIKV (MOI 1 or 5) and RNA was isolated 24 h later. Levels of *IFIT1* and *MX1* mRNAs were determined with RT-qPCR and C_T_ values normalized to *GAPDH*; (**G**) A549 cells were infected with ZIKV (MOI 0.1) and RNA was isolated at the indicated time points. RT-qPCR was performed and ZIKV RNA levels are presented relative to *GAPDH*. Data in **A**, **C** and **E** are representative of two independent experiments. Data in **D**, **F** and **G** are pooled from three (**D**,**G**) and four (**F**) independent experiments. Each data point is the mean value of two technical replicates. Statistical analysis: One-way (**D**) and two-way (**F**,**G**) ANOVA with Tukey’s multiple comparison (* *p* < 0.05, *** *p* < 0.001).

**Figure 2 cells-09-01476-f002:**
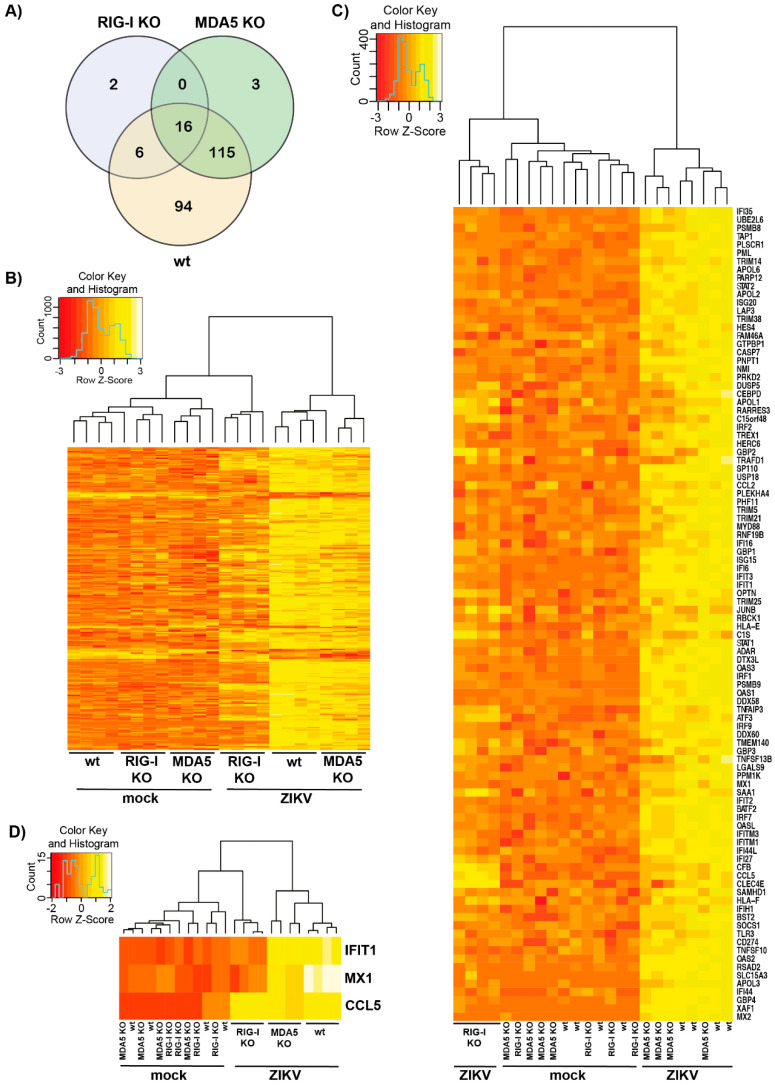
RIG-I sensing drives most transcriptional changes upon ZIKV infection. (**A**) Venn diagram showing differentially expressed genes in wt, RIG-I KO and MDA5 KO cells 24 h after infection; (**B**–**D**) Heat maps depicting all 236 differentially expressed genes (**B**), 98 differentially expressed ISGs (**C**) and three exemplary ISGs with different expression profiles in A549 wt, RIG-I KO and MDA5 KO cells upon ZIKV infection (**D**). Colors represent *z*-scores that indicate a value’s relationship to the mean, measured as standard deviations from the mean. *z*-scores calculated for each row (i.e., each gene) and were plotted instead of the normalized expression values to ensure that expression patterns are not overwhelmed by absolute expression values. Data in (**A**–**D**) pooled from four independent biologic samples.

**Figure 3 cells-09-01476-f003:**
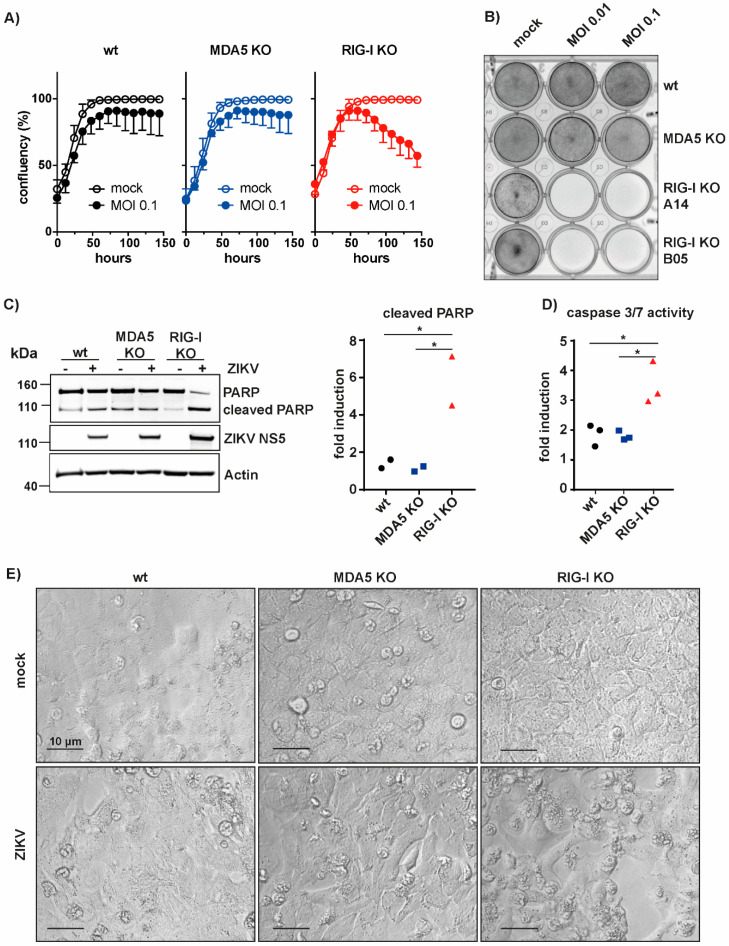
RIG-I signaling protects A549 cells from ZIKV-induced cell death. (**A**) A549 cells were infected with ZIKV (MOI 0.1) and cell confluency was measured for 6 days in the Incucyte; (**B**) A549 cells were infected with ZIKV, fixed 6 days after infection and stained with crystal violet; (**C**) A549 cells were infected with ZIKV (MOI 0.1) and lysed 4 days after infection. Cell lysates were analyzed by western blot using the indicated antibodies (left). Actin served as a loading control. Signal intensity of full length and cleaved PARP was quantified relative to background and the fold induction of cleaved PARP was calculated (right); (**D**) A549 cells were infected with ZIKV (MOI 0.1) and the activity of caspase 3 and 7 was determined 4 days after infection using the Promega Caspase-3/7 Glo assay; (**E**) A549 cells were infected with ZIKV (MOI 0.1) and images were acquired 4 days after infection. Scale bar: 10 μm. Data in **A**, **C** (right) and **D** are pooled form two (**C**) or three (**A**,**D**) independent experiments. In **A**, mean and SD are plotted; in **C** and **D**, data points correspond to individual experiments and statistical analysis was with one-way ANOVA with Tukey’s multiple comparison (* *p* < 0.05). Data in **B**, **C** (left) and **E** are representative of two (**C**) or three independent experiments (**B**,**E**).

**Figure 4 cells-09-01476-f004:**
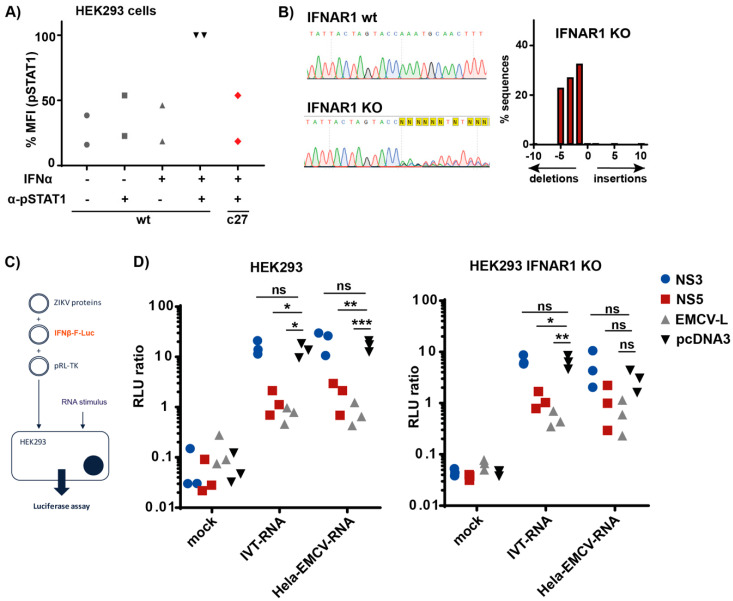
ZIKV NS5 inhibits IFN induction. (**A**) wt HEK293 and IFNAR1 KO clone c27 cells were stimulated with 5000 U/mL IFNα2a for 30 min before fixation and intracellular staining with α-pSTAT1 antibody. As controls, some wt cells were not stimulated or the α-pSTAT1 antibody was omitted. pSTAT1 levels were determined by flow cytometry. Mean fluorescence intensity (MFI) of the pSTAT1 signal was calculated and set to 100 in wt cells; (**B**) genomic DNA was extracted from IFNAR1 KO clone c27 cells. A fragment of DNA surrounding the targeted area was amplified by PCR and sequenced (left). Sequences were analyzed using TIDE (right). Number of nucleotides inserted or deleted, and the percentage of sequences affected are shown; (**C**) schematic of the experiment in **D**; (**D**) wt and IFNAR1 KO cells were transfected with the indicated expression plasmids, a plasmid encoding F-Luc under the control of the *IFNβ* promoter and a plasmid, which expresses R-Luc. Twenty-four hours later, cells were transfected with 5 ng IVT–RNA or 50 ng Hela–EMCV–RNA per well. F-Luc activity was determined 24 h after RNA transfection and normalized to R-Luc. Data in **A** and **D** are pooled from two and three independent experiments, respectively. Data points correspond to individual experiments and in **D** are mean values of technical triplicates. Statistical analysis: Two-way ANOVA with Tukey’s multiple comparison (ns *p* ≥ 0.05, * *p* < 0.05, ** *p* < 0.01, *** *p* < 0.001).

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
