# Peer review of "RIG-I Plays a Dominant Role in the Induction of Transcriptional Changes in Zika Virus-Infected Cells, which Protect from Virus-Induced Cell Death"

_cells, 2020, doi:10.3390/cells9061476_

Round 1

Reviewer 1 Report

Manuscript ID: cells-820904
Schilling and co-authors have resubmitted a revised manuscript “RIG-I plays a dominant role in the induction of transcriptional changes in Zika virus-infected cells and protects from virus-induced cell death”  to the Special Issue on Zika Virus and Host Interactions.

The authors have provided new data to confirm their results and interpretations of these results. The authors have responded to the issues raised on previous submission and have re-written parts of the manuscript that raised concerns.

The resubmitted manuscript addresses all the points raised by this reviewer and is thus acceptable for publication.

Only one minor addition is required: Please indicate at Materials and methods/Generation of knock-out cells/line 74: A549 and HEK293 cells

Reviewer 2 Report

The authors have correctly answered to my remarks. 

Reviewer 3 Report

The study by Schilling et al. very nicely show that in A549 cells RIG-I plays a major role in sensing and response to ZIKV. All in all, the data is conclusive, experiments well done. In the first part, they conclusively show that RIG-I senses ZIKV and induces transcriptional changes. In the second part they show that absence of Rig-I leads to increased cell death over time. However, it is a little blurry whether authors present the data as it is a direct effect of a simple consequence of lack of sensing and immune response which results in more replication, and thus more viral induced cell death. In This context the experiments are simply another way of showing that RIG-I is important in sensing and restricting ZIKV replication, strengthening their initial findings. It is a fine difference but should be made clear. Therefore, the statement in the title that RIG-I protects from cell death, or in the text from apoptosis is misleading, as direct interference of RIG-I with apoptosis are not shown. Finally, the last part where they show that NS5 might interfere with RIG-I signaling is interesting, underlines the importance of RIG-I in this setting and opens the way for future studies.

Change title: To make the claim that RIG-I protects cells from ZIKV-induced cell death more mechanistic data are necessary. RIG-I probably results in less viral replication leading to less cell death.

89: if these cells are optimized for virus growth, why not use them for ZIKV propagation?

118: usually abbreviated fwd

140 and throughout: separate numbers from units by space

163: if data was produced I always prefer that they are shown

Fig 1B: why is there no unstimulated control? Data are valid, just for consistency and curiosity of the reader

Fig 1: all in all very nice and conclusive experiments. However, there are two experiments that would add to the finding. First, determining of the infection rates of the cells with a more quantitative readout instead of western blotting. Secondly, adding supernatants to cells might lead to infection of the cells by progeny virus (even though 293s are not very susceptible). Could this be excluded by inactivation of the supernatants e.g. by UV irradiation or molecular weight filtration to retain virions

219 rephrase rather instead

233 RIG-I “sensing” protects from apoptosis. It is not shown that RIG-I itself is linked to apoptosis (as it is also stated by the authors in the discussion, but this should be made clear throughout the manuscript, title, results, etc.)

251 This is very likely the case but would be nice to see data showing kinetics of the viral titers in the supernatants.

Fig 3. The weakness of this part is that RIG-I does not directly affect apoptosis. Like the authors state it is probably reduced viral replication, leading to less apoptosis (306). However, the title suggests that RIG-I protects cells from apoptosis, but this is not a direct effect but an effect any factor restricting viral replication would have. If ZIKV is not sensed, virus replicates and cells die. The reverse conclusion that RIG-I protects from apoptosis cannot be made. Make this clear. Other than that the data nicely add to the previous finding that RIG-I sensing is crucial in limiting ZIKV replication.

306 provide reference

315 is this really comparable? Same MOIs?

309 how is the sensing and RIG-I levels in neuronal cells, which authors argue ZIKV leads to apoptosis probably causing microcephaly. Maybe discuss other cells where RIG-I levels and ZIKV apoptosis are known? Or suggest to look at this.

Fig 3 Please check: are the viral loads higher in supernatants from Rig-I positive or negative cells?

Author Response

This manuscript is a resubmission of an earlier submission. The following is a list of the peer review reports and author responses from that submission.

Round 1

Reviewer 1 Report

The manuscript “Virus sensing by RIG-I protects from Zika virus-induced cell death” by Mirjam Schilling, Nicki Gray, Jonny Hertzog, Philip Hublitz, Alain Kohl and Jan Rehwinkel was submitted to the Special Issue “Zika Virus and Host Interactions” of Cells.

In this manuscript, the authors have generated A549 cell lines with RIG-I, MDA5 or IFNAR1 gene knock-outs. These cell lines were used to study the effects of RIG-I and MDA5 pathways on Zika virus infection.

The manuscript nicely confirms with new method (IFNAR1 knock-out cells) earlier published data that Zika virus is able to inhibit RIG-I pathway, and not just IFN-induced pathway. And that this inhibition is (at least partly) due to NS5 protein. Also, the manuscript indicates that RIG-I, but not MDA5, is the major pattern recognition receptor in Zika virus infection, and that infection, especially infection of RIG-I KO cells, results in apoptosis. However, apart from very useful cell lines generated and transcriptomics assay (the last for which this reviewer has major concerns, see below) there is no other novel findings in this manuscript. This manuscript does not fulfill the scope of Cells in the depth required.

Major concerns:

The title of the manuscript implies that RIG-I protects from virus-induced cell death. The data presented does not support this conclusion. See point 5 below.
Lines 91-93 on the Materials and Methods and Figure 1C-E: for Figure 1C the medium obtained from the cells and used in the type I IFN assay, was the virus in the medium inactivated? If there was infectious Zika virus present in the medium, could this affect the results? In Figure 1C where medium from RIG-I KO cells does not induce ISRE-luc, is this due to medium lacking IFNs or could the infectious virus block IFN-induced pathway (as the virus is known to do)? Especially this is of concern since in Fig. 1E it is shown that RIG-I KO cells produce 100-fold more viruses. Thus, even though the initial MOI in the experiment for Fig. 1C is the same, within the incubation time (48h) the RIG-I KO cells could produce considerably higher amount of viruses and this could affect the outcome of the assays on the reporter cells. Was the MOI/titer determined at the point of induction/infection of reporter cell line? To rule out the effect of infectious virus in the medium used in IFN-assays, the virus should be inactivated (or titered at the time of measurement). The same concern applies to Fig. 1D experiment: Could the lack of IFIT1 and Mx1 induction be due to higher amount of Zika virus produced in RIG-I KO cells and the virus would inhibit the induction of the ISGs?
Line 178: the conclusion that ISGs are regulated in a RIG-I and MDA5-dependent manner is misleading. It implies here that RIG-I and MDA5 directly regulate ISGs, for which the data gives no support. Rather, the IFNs produced by RIG-I and MDA5 pathways regulate ISG-expression, the fact that is already known. As authors write on lines 172-173, this experiment was done “to examine the impact of reduced type I IFN secretion on ISG induction”, and they confirm the existing knowledge that less IFNs leads to lesser induction of ISGs.
Lines 205-208: conclusion that RIG-I induce ISGs in Zika virus infected cells is not supported by the results. When the cells are infected with Zika virus (MOI 5) and analyzed at 24hpi, is it assumed (or measured) that the cells produce IFNs? Based on Fig. 1C by 48hpi at least MDA5 KO and wt cells are producing IFNs. Could this observed ISG induction then be due to secreted IFNs, and not due to RIG-I or MDA5 directly? Also, if RIG-I KO cells produce less IFNs, then it is obvious that less ISGs are expressed in the infected RIG-I KO cells due to lesser amount of produced IFNs. Hence the observed phenomenon that RIG-I KO cells express less ISGs and the conclusion that the ISG expression is thus dependent on RIG-I is not supported and is misleading. Rather it is that due to low IFN production from RIG-I KO cells the ISG activation is low – a well-known phenomenon.
Lines 210-227: This reviewer has major concern about this experiment. As pointed out by the authors, it is known that Zika virus infection causes apoptosis. To examine the role of RIG-I and MDA5 in the course of apoptosis, the authors infected the KO cells and monitored cell death and apoptosis. Whereas the methods used for the general observation of cell death and apoptosis are adequate, the results do not support the main conclusion that RIG-I would protect from apoptosis. Nor do the results support statement on lines 23-24 on the Abstract that “loss of RIG-I and consecutive type I interferon (IFN) production causes virus-induced apoptosis”. The main concern here is that, as shown in Fig. 1E, RIG-I KO cells produce 100-fold more viruses within this time period. The considerably higher amount of infectious viruses would lead to observed apoptosis and faster cell death in RIG-I KO cells. Was the virus amount titered at the time of analyses? This observed phenomenon could be due to following: RIG-I KO cells do not produce IFNs because of the lack of RIG-I pathway -> cells are not protected from infection because less IFNs leads to lesser induction of ISGs -> this results in faster and higher replication of virus -> this in turn results in faster apoptosis and cell death. Thus, it is not the RIG-I directly affecting apoptosis, rather it is the lack of RIG-I affecting the innate immune responses of cells and making these “immune deficient” cells more prone to infection (and hence apoptosis). On lines 290-292 on Discussion the authors also point this out.
The above-mentioned concerns also apply for the conclusions on Discussion: lines 266-267 (ISG induction RIG-I dependent or RIG-I-activated IFN-dependent?) and lines 279-281 (apoptosis due to loss of RIG-I or higher virus replication?).

Minor points:

Lines 95-100: indicate the manufacturer of the luciferase assay kit and the plates that were used in this study. Lines 117-118: Life Technologies with capitals Figure 2C: the list of gene names on the side of the heat map are not visible. A separate list should be provided.

Reviewer 2 Report

This a manuscript by Schilling and al. which claim that RIG-I is the main sensor for Zika virus (ZIKV) in A549 cells. The authors showed that a loss of RIG-I causes a more pronounced ZIKV-induced apoptosis in these cells. The first important limitation of the manuscript is the lack of a clear rationale for the use of low or high multiplicities of infection (m.o.i.) of ZIKV through the various experiments. The susceptibility of the host-cell to ZIKV infection can greatly vary according to the viral doses used. The last one concerns the two independent parts of the manuscript showing importance of RIG-I in ZIKV-induced apoptosis and then the critical role of NS5 in inhibition of IFN induction as it has been already demonstrated elsewhere. The rationale for a such grouping is rather unclear for the reader. The last comment is that the content of the manuscript is much broad that the title covers. Consequently, the title should be changed to be more informative.

Specific comments:

#1. In Fig. 1, the authors need to explain why the 1 m.o.i was used for infection of A549 cells in graphs 1 CD and 0.1 in graph 1 E (and also 5 m.o.i in Fig. 2). The kinetics of viral growth in cells infected with ZIKV and host-cell responses to virus infection vary according to the m.o.i used. A low m.o.i., multiple cycles of infection can occur whereas a single round of infection is usually observed at high m.o.i. Consequently, it should be informative to determine the viral growth in RIG-I KO cells infected with ZIKV at 1 m.o.i. Only the quantification of viral RNA by RT-qPCR was performed on intracellular RNA collected from ZIKV-infected cells. It is of interest to quantify the amounts of virus particles released from infected RIG-I KO cells and also evaluate their infectivity by plaque-forming assay on Vero cells.

#2. In Fig. 2, a supplementary table listing the 98 differentially expressed ISGs in RIG-I KO cells infected with ZIKV is required. Too little information on ISGs such as viperin, a potent inhibitor of ZIKV replication, is provided by the authors except IFIT-1 and MX1.

#3. The Fig. 3 is unreadable and thus, the analysis of data was strongly constrained. As it has been stated above, it is unclear why the authors used only very low to low m.o.i for the monitoring of ZIKV-induced apoptosis in infected KO cells.

#4. Lines 285-86. Bos et al. (Virology 516: 265, 2018) reported that infection of A549 cells with ZIKV at 1 m.o.i caused apoptosis at 24 h (historical African strain) or 48 h (epidemic Brazilian strain) post-infection. It would be interesting to assess the susceptibility of RIG-I KO cells to infection with historical African strain of ZIKV.

#5. Line 279. The authors conclude that RIG-I prevents ZIKV-induced apoptosis in A549 cells. Recently, Turpin et al. (Cells, 2019, 8, 1338) demonstrated a role for Bcl-2 family proteins in the ability of ZIKV to control virally-mediated apoptosis. It would be of interest to discuss the possibility of an “orchestration” between RIG-I and Bcl-2 proteins in the control of apoptotic pathway by ZIKV though the action of its nonstructural proteins.

Reviewer 3 Report

2.1. cells lines and viruses: i suggest do describe the knock out techniques in a separate chapter. Potentially virus propagation as well.

Vero and A549 BVDV NPro cell origin and culture condition is not described.

2.2. what does the plasmid encode? which substrate was used? Were the cells lysed? What is the procedure?

Fig 1 A: why is there no – IFNA/D control for RIG-I wt?

Fig 1C: Can you please show data on the cell viability. If the cells are dying upon infection, this could also result in less IFN released.

Figure 1D: Capitalize MOI

Figure 1E: Infection at a low MOI and measuring replication is a very nice experiment. Here also analyzing IFN secretion and ISG induction would be interesting.

Line 115: typo: using the using the

Line 160: why did you chose RIG-I KO clone B05?

Line 179: was the abbreviation RLRs introduced before?

Line 182: Very general statement, might be different in other cell types

3.2. A title indicating the result would be consistent to the other titles

3.3. in parallel showing the ZIKV titer like in Fig 1E would be interesting. The experiments suggest that despite the cells dying, more virus is released.

Line 214: Somewhat surprising that MOI 0.1 6 dpi does not lead to any significant loss of confluency in wt A549 – have you also done this with a slightly higher MOI?

Line 217, indicate that this was done after 6 days

Figure 3B: Capitalize MOI

Fig 4 Please also determine the IFN secretion using the IFN bioassay (2.2)

Line 263: Again a very generalized statement, especially as you later also note cell-type specific differences with regards to sensor expression and co-factors

4. NS5 is present in all the assays where you infected with ZIKV. Please discuss this.

Can the authors comment on why specifically A549 cells were used? Other studies have already shown the importance of RIG-1 for ZIKV sensing in HEK293T (Chazal et al. Cell Reports 2018), dendritic cells (Bowen et al. Plos Path 2017) and skin cells (Hamel et al. J Virol 2015), as also cited by the authors.